# Building a COVID-Safe Navigation App Using a Meta-Model Based Context Server

**DOI:** 10.3390/s22249890

**Published:** 2022-12-15

**Authors:** Manfred Wojciechowski, Patrick Pogscheba

**Affiliations:** Faculty of Media, University of Applied Sciences Duesseldorf, 40476 Duesseldorf, Germany

**Keywords:** context awareness, internet of things, development methodology, meta model, context query language

## Abstract

Building context-aware applications is an already widely researched topic. It is our belief that context awareness has the potential to supplement the Internet of Things, when a suitable methodology including supporting tools will ease the development of context-aware applications. We believe that a meta-model based approach can be key to achieving this goal. In this paper, we present our meta-model based methodology, which allows us to define and build application-specific context models and the integration of sensor data without any programming. We describe how that methodology is applied with the implementation of a relatively simple context-aware COVID-safe navigation app. The outcome showed that programmers with no experience in context-awareness were able to understand the concepts easily and were able to effectively use it after receiving a short training. Therefore, context-awareness is able to be implemented within a short amount of time. We conclude that this can also be the case for the development of other context-aware applications, which have the same context-awareness characteristics. We have also identified further optimization potential, which we will discuss at the conclusion of this article.

## 1. Introduction

Discussions on the development of context-aware applications, including technologies and methodologies, are not new. A lot of these discussions were carried out before the 1990s with Marc Weiser’s idea on ubiquitous computing [1] and is still ongoing. Dey [2] defined an application context-aware if: “it used context to provide relevant information and/or services to the user, where relevancy depends on the user’s task”. According to Dey [2], context is: “any information that can be used to characterize the situation of an entity”, where “an entity can be a person, place, or object that is considered relevant to the interaction between a user and an application, including the user and applications themselves”.

An example for a context-aware application is a context-aware tour guide, which provides information to their users depending on their location and/or personal preference [3]. Another example is the pre-registration of elevators in a smart building using the localization of passengers within the callable range of an elevator, which should reduce the waiting time for passengers [4]. Numerous such examples can be found in different application areas, such as health care [5], residences for the elderly [6] and industrial applications [7].

Context-aware applications aim to reduce the need for user intervention [8]. Such applications can provide an automatic context-triggered action [9] e.g., initiating an emergency call if a fall of an inhabitant living in a smart home has been detected.

With the development of the “Internet of Things”, several industries are experiencing a digital transformation, where the development and usage of a “Digital Twin” is considered a technology that can help gain competitive and economic advantage over its competitors [10]. A Digital Twin refers to the virtual copy or model of any physical entity, both of which are interconnected via exchange of data in real time. In a broader view, the virtual copy or Digital Shadow, which is updated through sensors, can be applied in context-aware applications [11]. In [12], the authors describe that future Digital Twins have to be context-aware.

Research on context-awareness has been conducted since the 1990s; the development of such applications is still very challenging and expensive. We need to simplify the development process, in order to help overcome high application overheads, social barriers associated with privacy and usability, and an imperfect understanding of the truly compelling use of context-awareness [13]. A very detailed overview on the context-awareness system engineering challenges and applied techniques is given in [14]. One of many challenges is the way context information is being handled. In order to simplify the development of context-aware applications, a separation is required on how context is acquired to how it is used. The way context information should be used in such applications is without implementing the knowledge of sensor details and their implementation [15]. According to Henricksen et al. [3], such context information can originate from four types of sources. They can be “sensed, static, user-supplied (profiled) “ or “derived”. Context-aware applications must build on such information, which may require additional interpretation to be significant for applications [16]. In reference [17], three different architectural approaches for the implementation of the context related functionality are identified: direct sensor access, middleware infrastructure, and context server. In references [18,19,20], overviews on existing context frameworks, servers, and a discussion of their features, are given. Another challenge is the methodology of building context-aware applications. The requirements elicitation process is necessary to understand the user’s needs. In reference [14] an overview of existing approaches are given. These requirements can be formally specified through different modeling approaches, e.g., the Context Modeling Language (CML) [21]. The resulting context model can then be implemented using different approaches, e.g., key-value models. An overview and discussion is given in [14,22]. Additionally, design evaluation can be applied to verify that the implementation of context-aware behavior has been carried out correctly. This can be conducted through model checking, e.g., reference [23]. All these aspects have to be included into a methodology for the development of context-aware applications. In reference [14], the desirable features for a methodology are identified. Additionally, existing methodologies and tools are identified and evaluated regarding the desirable features. As a conclusion of that evaluation, the authors state that there is a strong tool support for the design and development of context-aware applications.

Though a number of methodologies and tools exist and are well discussed in the research community, the challenge still exists. We need experiences from the application of those methodologies in real applications. Therefore, based on these discussions, we follow an inductive approach to the question of how to minimize the development overheads and simplify the implementation of context-aware applications. We have developed our methodology, which uses a meta-model based context server. A meta model defines the modeling concepts that can be used to describe concrete models [24]. We will implement applications using this methodology with different context-aware characteristics and analyze its implications. In this paper we will start that work and describe and analyze the implementation of a COVID-safe navigation app, which has relatively simple characteristics. 

Our paper is organized as follows. The next section describes the related work and places our approach in its context. In Section 3 we describe the characteristics of the COVID-safe navigation app. Then, we describe the basic concepts of the context server. Section 5 then describes the methodology. In Section 6 we describe the development of the COVID-safe navigation app using the methodology and the context server. We then discuss our results.

## 2. Related Works

As stated in the introduction, we will follow an inductive approach towards a methodology to build context-aware applications. Therefore, papers that describe the application of a methodology for the development of concrete context-aware applications are of interest. In reference [25] we have described an initial model-based approach to build context-aware applications in a smart home environment. From then until now, we have improved the meta-model, optimized the development process, and implemented a near real-time processing context-server with an intermediate context query language. These improvements are presented in this paper.

In [19] there is an interesting investigation on how to bring context-awareness into the internet of things. The authors investigated 50 context-aware projects and tried to identify the lessons that we can learn in the IoT perspective. There are many requirements that have been identified from investigating these projects, which a context-aware IoT infrastructure should support, which also is partially supported by our context-server. What is missing for our purposes are the implications on the support to ease the development process.

There are a number of model driven development approaches, which aim to specify a context-aware application including the context conditions and the application behavior via a high-level abstraction, which then can be transferred to its implementation. The description is based on a meta-model, which can be used to describe concrete applications. One such example is presented in [26]. The modeling language, ContextUML, allows us to model the context-aware behavior of a web service. It describes the retrieval of single context attributes and when and how to automatically execute or modify the call of a web service with parameters that represent the context. In contrast to our approach context is limited to context attributes that can be relevant to the instantiation of the web services. A very complex meta-model PervML is described in reference [27]. It includes different sub-models, which can be used to describe different aspects of a pervasive application. It includes a structural model, which can be used to describe a physical location in which a service is deployed. Additionally, a user model can be used to describe context data on users and policies. These parts of the meta-model are focused on the conditions and instantiation of the services. Our approach on the contrast is focused on describing such context attributes in its broader context regarding the states of entities and their relations. This is necessary if context information should be evaluated in the context of their surrounding environment, e.g., finding paths with given context conditions. We also do not model the application logic of a context-aware application, but leave this to the programmer. We provide a context query language, which the programmer can use to access the context model.

Work on a meta-model based approach, which is only focused on the context model, can be found in reference [28]. The authors describe the motivation for a model-based approach to develop context-aware applications. Their goal is to provide a methodology that is consistent based on the meta-model through design and run time. They have published the meta-model in reference [29], but further publications on the implementation of the approach and a proof of concept cannot be found.

## 3. Characteristics of the COVID-Safe Navigation App

An example of a simple context-aware application is what we have implemented into our COVID-safe navigation app. This app allows university members and students of the university of applied sciences in Duesseldorf, Germany, to identify and follow paths to a destination room within the campus buildings which are “corona-safe” according to the lowest density of people on the desired path. The user can select a destination room and will get a visualization of the building structure and the safe path s/he can use.

The application uses simple context information that is either static or easily sensed. Static information includes the structure of the university building. In this structure the construction of the building including the rooms, lecture halls, floors and their connections have to be described. The density of people in these areas have to be sensed. Such information can be provided by simple sensors. Small deviations or sensing errors may not necessarily lead to suboptimal path recommendations. Therefore, coping with the unreliability of sensors is not necessary. Additionally, we do not need the identity of the people who are located in these areas. Therefore, we do not have privacy issues, as long as we do not use tracking of the mobile devices as an input for the density calculation.

In reference [30], a good overview on requirements of location models can be found. A major requirement of our application is the support of a hybrid location model, which includes a symbolic and a geometrical model. We need a symbolic location model in order to define connections between areas from a starting to a destination room, which represent the possible paths. For this, the location model has to support navigation queries. In order to visualize the building and the safe path, but also for the calculation of the walking distance we need a geometric location model, which can be used for position queries. Both location model types have to be combined. Nearest neighbor and range queries are not required in our application. 

High-level context information that has to be derived from sophisticated sensors and algorithms, which could be prone to uncertainty, e.g., the situation, activity, or goal of a user, is also not required. The user has to explicitly input his destination. A derivation of the destination from her/his lecture schedule is not part of the application.

The navigation app uses the context information in order to visualize the structure of the building and the suggested path per user’s request. In this aspect the application can be classified as “Proximate Selection” [9]. The application provides information dependent on the context, which is manually entered by the user. Therefore, we do not have to deal with undesired or even harmful behavior. The user will always be in control of the application. However, we still have to deal with user acceptance in case that the suggested path is obvious nonsense or includes a detour that is not acceptable, even if it is the safest path. The context-awareness characteristics of our application is summarized in Table 1.

## 4. Basic Concepts of the Meta-Model Based Context Server

### 4.1. Context Server

We have implemented the HSD context server, which directly supports the meta-model as described in the next subsection. It allows us to connect sensor data to the context-model and to query the context model in near real-time. The core of the context server implements an in-memory extended graph database, where entities and relations are represented by nodes and edges. It allows the definition of node and edge types. Node types can be part of a multiple inheritance relation. A concrete graph, which represents the application specific context model, can be constructed based on the defined node and edge types.

Besides explicitly defined edges, implicit relations are also supported, which result from the application of comparison operations between edge properties. The graph database also directly supports geometric location data types and operations, which can also be part of an implicit relation.

### 4.2. Meta-Model

Based on a meta-model, the developer can define the application specific context model. An overview of the relevant part of the meta-model is given in the following Figure 1.

The meta-model allows us to define entity types with a set of attributes. An entity is any object in the real environment that is considered relevant for the specific context aware application. We can define abstract entity types and also a subtype-relation between entity types, where attributes are inherited from the super-types.

We can define relation types between two entity types. These relation types can be directed, meaning that the relation is only valid from the source entity type to the destination and not vice versa.

Attributes define the named properties of entities or relations and are of a defined data type, e.g., numeric. An attribute can have a default value.

All the model elements can have a description, which allows us to document the context model schema. 

### 4.3. Location Models

The context server supports both geometric and symbolic location models. A good overview on location models can be found in reference [30]. A geometric location model is based on geometric coordinates, such as a geometric location model, implemented in the context server by the special data type ‘location-geometric’. Currently, this data type supports the definition of a rectangular cuboid with a given starting point (x_start_, y_start_, z_start_) and an ending point (x_end_, y_end_, z_end_). This data type includes a distance function, which allows us to calculate the walking distance of a path, but also to execute range queries.

Additionally, the usage of a symbolic location model is supported. The meta-model allows us to define and use context entities as such symbols, e.g., the room name. Relation types can be used to define spatial relations, e.g., “containment” or “is connected”. The distance between two symbolic coordinates can be explicitly given in defined attributes of these relations.

The context server allows us to combine these different location models into a hybrid location model.

### 4.4. Context Query Language

The context server has an intermediate query language, which is very close to the implementation of the context server, and it is then directly compiled into the execution engine of the server. It is less developer friendly, but currently the only way to define queries that can be directly executed. The query language allows us to search and to traverse through the context model, which is internally implemented as a graph. The query language uses the JSON-syntax. The basic elements of the language are given in the following.

In its simplest form, the query language allows querying for entities of a given type with or without attribute conditions. The following Listing 1 shows the syntax of such an entity query. In Section 6.4 concrete examples of such queries are given.
**Listing 1.** Syntax of an entity query.{  "entityQuery": "<entity type>", "conditions": [  { "attribute": "<name>", "value": "<value>", "condition": "<operator>" },  ...  ] }

A relation query allows us to find specified relations between entities, which can be further reduced by entity queries. The following listing shows the syntax of a relation query. Relation queries can be further concatenated. Either the starting entity or the destination entity of a relation can be the start or the destination of another concatenated relation.

In order to traverse through the context model, a transitive relation query can be used. This means that if an entity “A” is related to an entity “B”, and an entity “B” is related to an entity “C”, then entity “A” is also related to entity “C”. Such a query can be used, for example, to find a path using a symbolic location model where symbolic locations are “connected” with each other. 

## 5. Development Methodology

Our development methodology consists of five steps. These will be outlined in the following. How we applied this methodology to implement the COVID-safe navigation app is described in the next section.

Step 1 is to define the application specific context model and to implement it into the context server using the meta-model. First, the conceptual context model is defined. The entities and relations and their properties are defined which are relevant for the application. Then, the identified entities are examined regarding common properties or semantics. These may be a reason for defining super entities that can be used to organize those entities in an object-oriented manner. The conceptual context model is then implemented into the context server using an XML-description. After the description is implemented, the entity and relation types and their properties are available in the context server. 

Step 2 is the instantiation of the context model. In this step, the concrete entities and relations are registered in the context server. The context server provides two kinds of interfaces for the registration. One possibility is to use a REST-interface to add, modify, and delete concrete entities and relations. As an alternative, one can use a JAVA-API. After this step, the context model is built internally in the form of an extended graph database.

The application of step 1 and step 2 in the development of the COVID-safe navigation app is described in detail in Section 6.3.

Step 3 is the assignation of the sensors to attributes of entities or relations of the model instances. This is conducted by using a configuration file. A “fingerprint” is used to associate sensor data to the properties of certain entities or relations. Incoming sensor data will then automatically update the context model. The application of this step in the project is described in Section 6.5.

After these three steps, the context server provides a digital environment model for the specific application. The following two steps will integrate the context logic into the application.

In step 4, the context model can now be queried and supervised using the context query language. The concrete queries will be defined in the next step and they can be manually tested using a web interface. The context queries for our project are described in Section 6.4.

In step 5, the application is connected to the context server using the REST-API. Through this API the application can execute context queries, and also subscribe to context change events. 

## 6. Development of the “COVID-Safe Navigation App”

### 6.1. Project Description

The HSD context server, together with its development methodology, was applied in the scope of a student project at the University of Applied Sciences, Duesseldorf, with students from bachelor’s and master’s courses in media informatics during the COVID-19 pandemic in 2021. The aim was the development of a context aware application which could be useful in the pandemic situation. The participants agreed to develop a COVID-safe indoor navigation app, which allows students to get from class to class while trying to obtain as little contact as possible. The project team consisted of three students, two masters and one bachelor student, all with some programming skills, but no experience in context awareness. The goal was to first gain basic knowledge in context awareness and then to use and configure the provided context server with a sensor infrastructure already implemented in the laboratory, and finally to build a prototype of the COVID-safe navigation app. Each student spent 16 days achieving these goals, which totaled the students’ workload to 48 days. 

### 6.2. Architecture

For better understanding we gave a rough, simplified overview of our architecture (Figure 2), which the application was built on. A sensor layer based on home/building automation systems (OpenHAB in our case) connects sensors and provides collected sensor data over MQTT, which is often used in IoT-environments, provided by a MQTT-Broker in JSON-format. The data is simple and provides the sensors’ fingerprint for easy integration and mapping. The central context layer contains the context server with its entity registration, context model and processing pipeline. The server connects to a MQTT broker and gathers “sensed” data from the sensor layer which can, if desired, be processed for “derived” data input to the context processing. Registered entities and their relations, e.g., rooms and their connections, are injected into the graph-based context model. A REST-based API is provided for querying the context model.

Arbitrary applications are possible: our application layer contains an Android-based application, which queries the context model and the entity Registry for the defined use case.

The application was built as a distributed networking system. This consisted of the building bound HSD context server, which used the building specific sensor information. These were provided through MQTT from the sensor infrastructure, based on OpenHAB (www.openhab.org, accessed on 13 December 2022). This approach leads to a high level of decoupling between sensor information and context generation to improve interoperability with different systems. In the context server, all entities and relations, e.g., rooms and their connections, are instantiated and registered for use in the context model. Low-level sensor data from the sensor infrastructure (“sensed” data) can be processed, if desired, into high-level (“derived”) data inside the context-servers extensible processing pipeline. APIs are introduced to query the context-model and the registered entities which are needed in the application layer.

The COVID-safe navigation app was implemented as an Android application. It used its own local HSD context server, which was running as a background service. The local context server uses a service discovery approach to find a remote context server. Then, it syncs the context models from the remote context server for privacy reasons and faster processing. This allows for an easy implementation of smart context spaces. Communication between local and remote context servers is done by REST-API-Calls. The COVID-safe navigation app then queries the local context server for the needed context information, e.g., the safe path to the class room. The distributed context-server approach introduces a high level of privacy and security as no information from the user is processed remotely.

### 6.3. Context Model

In the COVID-safe navigation app, the context model is used to visualize the interior structure of the buildings at the university. Students then receive an overview of the university and then to show the user a proposed route to the destination. The context model is also used to calculate the “safe” path with the least population density to the specified destination.

In the following, we will identify those elements of the context model that are needed for the visualization of the building structure. The building consists of a number of floors. Each floor can have a number of rooms, lecture halls, doors, corridors, staircases, and elevators, for which a name and the geometric location is given. A relation “has” relates to the floors in a building. We define a relation “is in”, which describes the parts that are inside a floor.

For the route calculation we need to describe which rooms, lecture halls, doors, corridors, staircases, and elevators are directly connected with one another. Using a “connected to”-relation we can then traverse from a starting location to a destination following all those parts which are connected with one another. In order to simplify the context query, we define an entity type “Area”, which is the super type for the entity types “Room”, “Lecture hall”, “Door”, “Corridor”, “Staircase” and “Elevator”. Each “Area” has a geometric location, but also an attribute “size” and “person density”. 

For all entity types that can be localized we define an abstract entity type “Localizable Entity”, which holds the position information. Such localizable entities are the “Building”, the “Floor” and the “Area”. Finally, we will define an abstract root entity type “Context Entity”, which allows us to query for any context entities and which holds the entity’s name. The following Figure 3 gives an overview on the entity and relation types that define the schema of the context model for the COVID-safe navigation app.

The above described conceptual context model schema is imported into the context server using an XML-description. The following Listing 2 is part of that description and defines the entity type “Localizable Entity”.
**Listing 2.** Importable context model schema description.<EntityTypename=**"LocalizableEntity"**> <generalizable>  <isAbstract>**true**</isAbstract>  <parentElementName>**ContextEntity**</parentElementName> </generalizable> <attribute name=**"position"**><type>**location-geometric**</type></attribute> </EntityType>

After the model schema is imported, the described context entity and relation types are then available in the context server. Finally, to complete the context model, the concrete entities and relations have to be instantiated. The following Listing 3 describes how to register a concrete entity using the Java-API of the context server. In this example, a new room is registered with its name, size, and position. The fingerprint information is needed for the sensor registration. After the instantiation, the final context model consists of 422 registered entities and 443 relations, describing the building structure at the university.
**Listing 3.** Instantiation of the context model.EntitySchema Room **=** thisEntityServer**.**getEntitySchema**()** **.**getEntityTypeManager**().**getEntityType**(**"Room"**)****;** Entity entityRoom04_E_007 **=**
**new** Entity**(**Room**)**; entityRoom04_E_007**.**setName**(**"04.E.007"**);** entityRoom04_E_007**.**setFingerprint**(**"04.E.007"**);** entityRoom04_E_007**.**setAttribute**(**"size"**,**
**new** AttributeNumber**(**"334.41"**));** entityRoom04_E_007**.**setAttribute**(**"position"**,** **new** AttributeLocation **(**"{\"xStart\":4.3, \"xEnd\":25.6, \"yStart\":20.3,\"yEnd\":36, \"zStart\":0,   \"zEnd\":0}"**));**  thisEntityServer****.****addEntity****(****entityRoom04_E_007**);**

### 6.4. Context Queries

For the COVID-safe navigation app, only two types of queries are needed. The first query returns all areas which are in a named floor. This can be achieved with the following Listing 4:
**Listing 4.** Retrieve all areas which are within floor “1”.
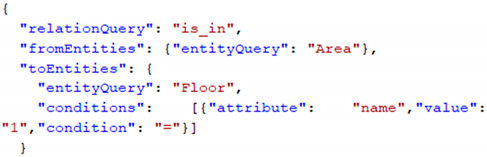


As a result, all registered area (sub-)entities within the floor “1” will be returned. If only specific areas should be retrieved, e.g., only rooms, then the first entity query can be replaced by “fromEntities”:{“entityQuery”:”Room”}.

The second query searches for paths from a starting area to a specified destination area and then sorts the results according to the aggregated person density of the included areas and also aggregates the walking distance, see the following Listing 5:
**Listing 5.** Find the paths from room “04.E.001” to the room “04.E.032” sorted by the person density.{
  "relationQuery"
: 
"connected_to"
,
  "fromEntities"
: { 
"entityQuery"
: 
"Room"
,
"name"
: 
"04.E.001"
},
  "toEntities"
: { 
"entityQuery"
: 
"Area"
 },
  "appendQueries"
: [{
   "relation"
: {
    "transitiveRelation"
:       { 
"relationQuery"
: 
"connected_to"
, 
       "toEntities"
: { 
"entityQuery"
: 
"Area"
 } },
    "appendType"
: 
"TO_WITH_FROM"
,
    "minDepth"
: 
0
    }, 
"appendType"
: 
"TO_WITH_FROM"
 }, {
   "relation"
: {
    "relationQuery"
: 
"connected_to"
,
    "fromEntities"
: {
"entityQuery"
: 
"Room"
,
"name"
: 
"04.E.032"
 },
    "appendType"
: 
"TO_WITH_FROM"
   } } ],
  "aggregation"
: [    { 
"name"
: 
"personDensity"
, 
"order"
: 
"asc"
 },    { 
"name"
: 
"distance"
 }   ] }


The first part of that query looks for all areas that are connected to the starting room “04.E.001”. The appended query then looks for all areas that are connected to the identified connected areas. All the resulting areas are then finally checked to see if they are connected to the destination room “04.E.032”. Then, each of the identified paths are aggregated to the accumulated person density in each area of the path and then the accumulated distance between the connected areas. The paths are then sorted regarding the population density. 

### 6.5. Sensors

For obtaining a measure of the occupancy in an area of a building, e.g., a hallway or a room, we need sensors for counting people currently in the areas that the users will be passing through. Besides (depth-)camera based sensors, there are also infrared sensors or light barriers for solving this problem. For our prototype we used a double infrared passage sensor with direction recognition from Homematic (https://homematic-ip.com, accessed on 13 December 2022) so we obtained inbound and outbound traffic. The sensor is integrated in our OpenHAB backend and mapped to an area b. Sensor updates, together with the fingerprint of the area, are sent to the context server in JSON over MQTT for integration into the context model.

A big advantage of using MQTT in the distribution of sensor information is good testability. Injecting arbitrary messages into the system is quite easy from any MQTT-client. Manual and automatic tests can be done by sending test messages and monitoring the application’s outcome.

The population density is then calculated in the context server pipeline by all persons currently sensed in an area of the building by means of entering and exiting information of the sensor divided by the bounded area size.

For testing purposes, we covered a small area in the ground floor of building 4 at the HSD. Some hallways and rooms were equipped with passage sensors at their boundaries. For accessing the sensors, a Homematic gateway was installed on a Raspberry Pi 4 together with OpenHAB, a WLAN access point software, MQTT broker and the context server service, everything containerized for quick deployment on arbitrary systems. The test system was placed near our testing area on the floor of Figure 4.

### 6.6. Application

The concrete application for COVID-safe indoor navigation was implemented as a mobile Android application. The main view shows a floor of the university building. All information about the building structure was obtained from the context server and then used for the drawing. The user can choose different floors. By tapping into rooms, hallways or elevators, the user then selects the current position and also the desired destination. An automatic localization of the user within the local context server was not implemented at that time. When submitting the navigation request the best path regarding minimum population density and distance is requested from the local context server and then is drawn onto the floor map in Figure 5.

## 7. Discussion

We have described how to apply the methodology to build fairly simple context-aware applications, such as the COVID-safe navigation app, using a meta-model based context server. We have described the basic characteristics of that app in Section 3. Based on the implementation of the COVID-safe navigation app, we have shown that a small team of students, which had no previous experience in context awareness, were able to build a context-aware application with these characteristics within a very limited time frame. The meta-model and the methodology were easy to understand and apply. At this point, we do not have a direct comparison of our approach with a manual implementation to prove that it simplifies the development of such context-aware applications. However, the student project gives some indication that this may be the case. In another student project, we implemented a digital fire brigade plan. In that project, the fire brigade is supplied with a mobile front end, which shows the structure of the building, the state of fire doors, the location of detected smoke and areas with a population density. It also calculates a suitable path from any entrance to the detected smoke area. We had the same result in that project. We assume that this will be the case for the development of any other context-aware applications with the same characteristics. Such an application could be used in an industry 4.0 scenario. The context server can be used for the identification of free transportation units nearby and the navigation of these units depending on the characteristics and state of the production environment.

However, the project also showed some shortcomings, which we have to handle in the future. The instantiation of the context-model using the Java-API was very time consuming, since each entity and relation had to be programmed manually, which can cause errors in the programming. A visual model editor could be of great help to construct and visualize the context model. Such a tool will be one of our next steps for improvement. Additionally, a tool to import building information from existing building planning software into the context model could be added.

After the university building’s structure was initially instantiated, we tested the context model manually using the path finding query, which in some cases resulted in false results, because of the wrong instantiations. It was time consuming to find potential mistakes in the model, and to correct them. A model checking tool that could be used to find and visualize mistakes would be needed. This would not only include errors in the instantiation, but also an evaluation of the calculated paths in different simulated person densities.

The implementation has also shown that the ability of the context query language to aggregate on single entity or relation properties may not be sufficient. The current implementation of the context server will always prioritize the paths with the lowest person density, regardless of the walking length of the path compared to other alternatives. This optimization is at the moment delegated to the application, but which should be provided by the context server in the future.

Non-functional features of the context-server were proved by the implementation of the application. The context server was able to cope with the context model of the application, which included 422 entities and 443 relations, which resulted from the description of the university’s building structure. The implemented in memory graph database did not have a problem with the size of the context model, but also not with executing queries that required a flexible traversal through the model graph in order to identify the possible routes, calculate the walking distances, aggregate the person density on the route and to prioritize these routes. In our runtime environment, it took around 9 ms for the context server to identify the possible paths, which typically included the description of 33 path alternatives and their aggregations.

## Figures and Tables

**Figure 1 sensors-22-09890-f001:**
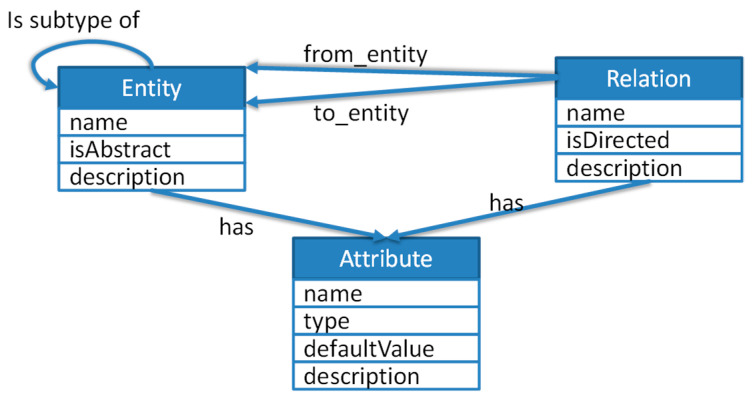
Excerpt of the meta-model.

**Figure 2 sensors-22-09890-f002:**
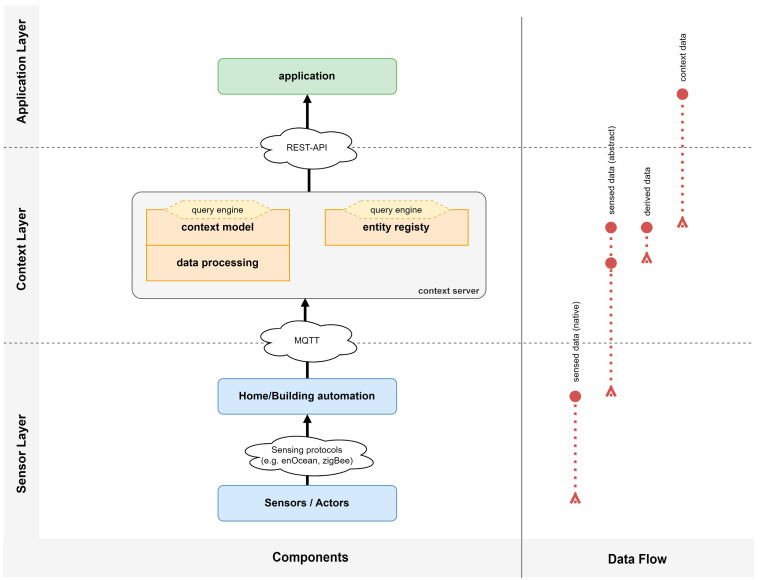
Architecture.

**Figure 3 sensors-22-09890-f003:**
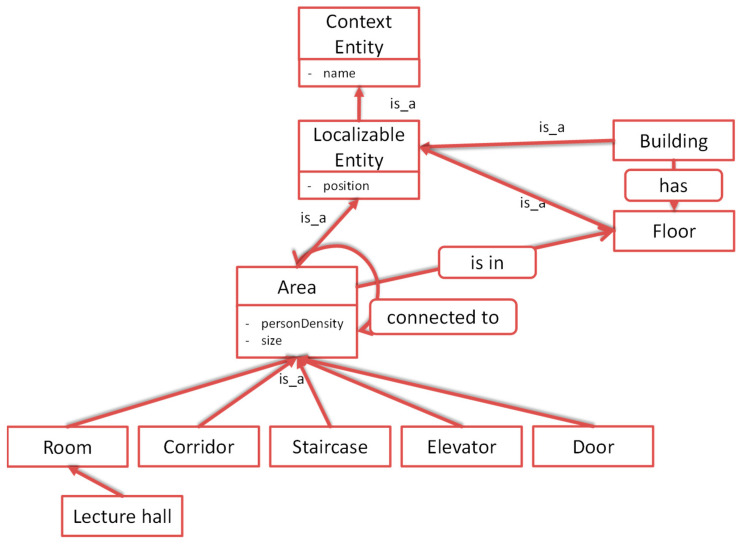
Conceptual context model of the COVID-safe navigation app.

**Figure 4 sensors-22-09890-f004:**
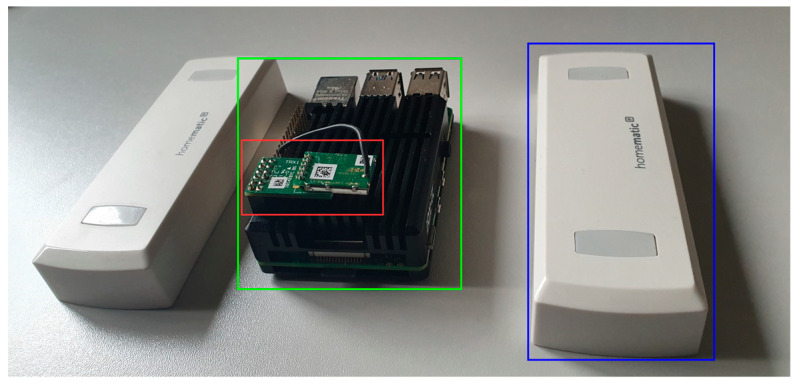
Used hardware in our test setup: Raspberry Pi4 (green), Homematic radio module (red), passage sensors (blue).

**Figure 5 sensors-22-09890-f005:**
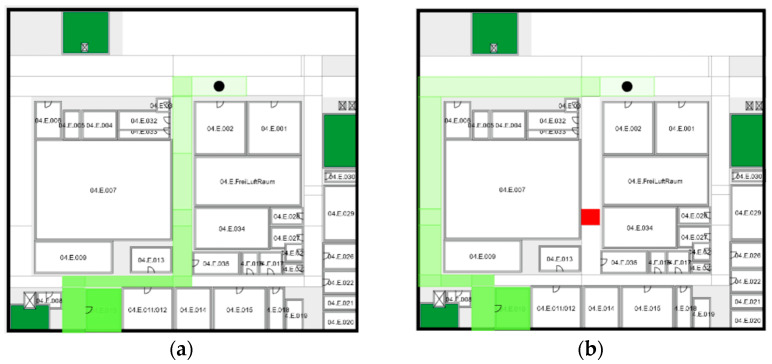
Navigation panel of the app without (**a**) and with areas of high person density (**b**) marked in red.

**Table 1 sensors-22-09890-t001:** Characteristics of the context-aware application.

Type of Context Information				General Aspects	
Static	sensed	profiled	derived	Privacy	acceptance
x	x	-	-	-	x
**Location Model**		**Location Queries**			
symbolic	geometric	position	navigation	nearest	range
x	x	x	x	-	-
**High-Level Context**				**Application Type**	
situation	activity	goal			
-	-	-		Proximate Selection	

## Data Availability

Not applicable.

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
