# Peer review of "Building a COVID-Safe Navigation App Using a Meta-Model Based Context Server"

_sensors, 2022, doi:10.3390/s22249890_

Round 1
Reviewer 1 Report
In this paper, authors build a meta-model-based system that allows to define and build 12 application-specific context models and integration of sensor data without any programming. They also use their system in a COVID-related navigation app.
The authors do a good job at describing their system and explaining its use. The system seems to be useful for different situations. The authors should have done a better job explaining the more generalizable uses of their system which would benefit more people in the field. Authors also lightly compare to the state-of-the-art; a more thorough comparison is needed. In terms of the writing, the paper is mostly fine, but the English needs to be revised. They should also refrain from showing raw code in the paper that doesn't really help the reader's understanding.
Author Response
Dear reviewer. Thank yor for your valuable input. We now give a better overview on related work. We relate our work to published model driven development approaches like ContextUML or PervML. Where the later is focused on describing the whole context-aware application on a high-abstraction level, our approach is focused solely on the context model. We have identified another model-based approach, which is also only focused on the context model. That approach is only described in theory with no implementation or proof of concept. In a new chapter we work out the context-aware characteristics of the COVID-navigation app. For the implementation of any other context-aware application with the same characteristics we assume that our methodology will be as efficient. We will remove some of the raw code where it is not neccessary.
Reviewer 2 Report
In this paper, authors proposed COVID-safe navigation app using a meta-model based context server. The idea is sound but the way the idea was investigated needs more improvement.
1. The research problem need to be highlighted in both abstract and introduction.
2. The organization of the introduction is not proper. As scientific research article, the organization of the introduction part should follow the research community writing style. Authors should first in introduce the problem well and motivate the readers to the importance of the problem. They should further navigate the literature and how people dealt with such problems in similar context. They should find the gaps and suggest their solutions before introducing the proposed approach as a solution. Towards the end of the introduction part, authors should detail out their contribution.
3. The state of art section should be renamed "related works".
4. The methodology should be presented clearly as steps, with sufficient details of every step
5. Figures are not clear, they should be enhanced.
6. The discussion part is very shallow. Authors should provide a case study to evaluate the proposed context aware approach.
7. As there is no evaluation benchmarking for such design, authors are at least required to test the functional and non-functional requirements of the proposed design.
Author Response
Dear reviewer. Thank yor for your valuable input. We have done a major revision on our paper.
Our research problem is now more clearly stated in the abstract and the introduction that we want to provide an efficient methodology to implement context-aware applications. To reach that goal we choose an inductive approach where we start with relative simple applications and investigate the suitablility of our meta-model based approach by the implementation of that application. In the beginning we identify the context-awareness characteristics of that application. We assume that for any context-aware application having the same characteristics the results can be applied also in their implementation.
We have revised the introduction according to the suggested structure. We have added more references to relevant papers.
We have renamed the chapter "State of the art" to "Related works"
In chapter 5 we give a coarse overview on the methodology. In chapter 6 we give a more detailed description on how to apply the steps of the methodology in the implementation of the application. We have added the references to the application of the steps within the overview in chapter 5.
We will rework on some figures and dismiss some.
We enhanced the discussion part toward the applicability of the methodology for context-aware applications with the same characteristics as the COVID-app. We have also better worked out the shortcomings we have learned from the project.
We have added some findings regarding the non-functional features of the implementation, which includes the size of the context model and the execution time for calculating the optimal route.
Reviewer 3 Report
The paper presents an important problem and is interesting, a few improvements are needed:
1) the introduction is concise which is good but there should be definitions and references to few words 'context', 'meta-based'. the definition of contexts can be picked from here https://doi.org/10.1016/j.cosrev.2019.01.001.
2) I think architecture figure 2 should be more professional , currently it looks like just using a tool.
3) the paper is interesting but it misses a flow of full research paper.
Authors need to have a proper section for proposed architecture, with flow in and out and describe each layer in a sequential order.
4) Authors need to define the scalability issues related to the application
5) Overall , paper look more like an assignment and does not come under rsearch paper section, is it a report? authors should also discuss about the open-source options related to this tool.
6) what is exactly the problem definition in this work? it should be mentioned explictly.
7) is this applicable using IOT services, focus should be more on high level architecture, the pieces of code can go to appendix or if they are in the paper, it should be under a subsection.
8) is there any evaluation technique to this method? I miss it in paper.
Author Response
Dear reviewer. Thank yor for your valuable input. We have done a major revision on our paper.
Our research problem is now more clearly stated in the abstract and the introduction that we want to provide an efficient methodology to implement context-aware applications. To reach that goal we choose an inductive approach where we start with relative simple applications and investigate the suitablility of our meta-model based approach by the implementation of that application. In the beginning we identify the context-awareness characteristics of that application. We assume that for any context-aware application having the same characteristics the results can be applied also in their implementation. The implementation of the COVID-safe navigation app is used as a proof of concept that the methodology can be applied and is suitable for applications with these context-aware characteristics. We try to evaluate this by letting a group of 3 unexperienced students implement this application. They have fully implemented the application by spending 48 days in total. We can only assume that without our methodology and tools they were not able to implement the application within a similiar time frame.
We have given a reference to the definition of context according to Dey.
We have given a definition and reference to "meta-model".
We will give a more in depth description of the architecture.
Round 2
Reviewer 2 Report
In this revised version, the paper still has some issues:
- Authors claimed that "the proposed methodology helps to reduce development complexity", however, there is no evidence of such claim in the paper.
- Figures are still of low resolution.
- English writing and style need more improvement.
Author Response
We have added to the conclusion that at this point we have not a direct comparison of our approach with a manual implementation to prove that and in which extend it simplifies the development of such context-aware applications. But the student project gives some indication that this may be the case.
We have replaced figure 2, 4 and 5 with a more high resolution version.
We have asked a native american (non academic) to improve the english writing and style.